

# Influence of rotor blade flexibility on the near wake behavior of the NREL 5MW wind turbine

Leo Höning[1,3], Laura J. Lukassen[1,2], Bernhard Stoevesandt[3], and Iván Herráez[4]

[1]Institute of Physics, University of Oldenburg, Carl von Ossietzky University Oldenburg, Küpkersweg 70, 26129 Oldenburg, Germany
[2]ForWind, Institute of Physics, Carl von Ossietzky University Oldenburg, Küpkersweg 70, 26129 Oldenburg, Germany
[3]Fraunhofer Institute for Wind Energy Systems – Fraunhofer IWES, Küpkersweg 70, 26129 Oldenburg, Germany
[4]University of Applied Sciences Emden/Leer, Constantiapl. 4, 26723 Emden, Germany

**Correspondence:** Leo Höning (leo.hoening@iwes.fraunhofer.de)

**Abstract.** High fidelity CFD simulations of the NREL 5MW wind turbine rotor are performed, comparing the aerodynamic behavior of flexible and rigid blades with respect to local blade quantities as well as the wake properties. The main focus has been set on rotational periodic quantities of blade loading and fluid velocity magnitudes in relation with the blade tip vortex trajectories describing the development of those quantities in the near wake. The results show that the turbine loading in a quasi-steady flow field are mainly influenced by blade deflections due to gravitation. Deforming blades change the aerodynamic behavior, which in turn influences the surrounding flow field, leading to non-uniform wake characteristics with respect to speed and shape.

## 1 Introduction

Since renewable energies gained increasing attention in the context of global warming, the wind energy industry has grown considerably over the last decades, even though the growth should be even larger to reach the climate goals (WindEurope (2022)). In this context, the ever increasing demand of energy, the urgency of replacing fossil resources and the economic demand for low prices in a highly competitive energy market lead to growing rotor sizes (Serrano-González et al. (2016), GWEC (2021)). Larger rotors allow for higher energy output and a reduction of the levelized cost of energy. On the other hand, increasing rotor diameters along with mass restrictions force the blade structures to become slender, which is accompanied by more flexibility and, thus, larger deformations of the blade due to wind loads. While blade flexibility has little effect on the overall rotor performance for smaller wind turbine rotors, aeroelasticity has significant effects on multi-megawatt turbines with long and slender blades (Rahimi et al. (2016)). Instabilities like vortex-induced vibrations or flutter occur with an increased probability and therefore gained increasing research interest over the last years (Horcas et al. (2022), Heinz et al. (2016), Heinz et al. (2016b), Manolas et al. (2022)). These highly unsteady phenomena need to be investigated further to ensure a prevention of unwanted blade vibrations. In this context, this work is intended to contribute to the basic understanding of deforming wind turbine blades.





Most wind turbine design tools are based on computationally efficient engineering tools using low-fidelity Blade Element Momentum (BEM) theory models. Low-fidelity models are often based on smaller, more rigid turbines and need to be corrected towards dynamic flow phenomena like dynamic stall for a rapidly changing angle of attack, 3D effects near the blade root and tip or skewed wakes for more flexible cases (Rahimi et al. (2018)). In order to gain more reliable results, high-fidelity fluid-structure interaction (FSI) tools that couple computational fluid dynamics (CFD) with computational structural dynamics (CSD) models have been recently receiving increasing attention.

Several studies have investigated the difference between rigid and flexible rotors with respect to aerodynamic loads. Yu et al. (2014) showed by using a non-linear Euler Bernoulli beam coupled once per rotor revolution to an in-house incompressible flow solver that the aeroelastic deformation of the NREL 5MW wind turbine with a rotor diameter of 126m results in a reduction of aerodynamic loads mainly due to torsional deformation. Similar findings were made by Dose et al. (2018), where the same rotor was simulated with OpenFOAM using a finite element method (FEM) based non-linear Geometrically Exact Beam Theory (GEBT) implementation including a loose coupling once per time step between the structure and the flow field that is calculated solving the Unsteady Reynolds-Average Navier-Stokes (URANS) equations, neglecting the influence of gravity. This study showed that the averaged aerodynamic power output is reduced on the one hand due to the blades bending towards a smaller rotor diameter and therefore less area to extract energy from and on the other hand twisting the blades towards lower angles of attack (AoA), which leads to a lower gliding ratio. This reduction of AoA due to the torsional deformation was also shown by Liu et al. (2019) for a floating setup under surge motion making use of OpenFOAM coupled to the multi body dynamic solver MBDyn. Guma et al. (2021) simulated a pre-bent 80m diameter turbine in turbulent inflow using the CFD code FLOWer, showing an increase in rotor torque for the flexible blades due to the bending resulting in an increasing AoA and a deformation towards a larger rotor disk area. The opposing behavior in these studies can be explained by the different structural models of the underlying turbines. However, these findings have not been related to effects in the respective wind turbine rotor wakes yet.

Besides the aerodynamic loading, wind turbine blades are influenced by gravity and centrifugal forces, that bend the blades independently of the aerodynamic forces. The influence of these forces is expected to be growing with increasing rotor sizes and the naturally accompanying flexibility of the structures. While the centrifugal force is expected to be constant in time for a constant rotational speed of the rotor and tends to straighten the blade in radial direction, the gravity force has a 1P blade passing frequency, with its direction depending on the azimuthal position. This leads to a dynamic blade deformation behavior in contrast to the almost constant influence of the centrifugal force. Despite its influence on deformation and loading, Sayed et al. (2019) concluded that its influence on the total power output of a generic 10 MW wind turbine is negligible. Its influence on the turbine wake has not been investigated in detail yet but it is important to understand the influence of blade flexibility in the context of wind farm layouts, where the wake of a turbine in the first row affects the inflow of turbines in the second row.

Comparisons between wakes of rigidly and flexibly modelled rotors have been studied for differently sized rotor blades. Under complex flow situations, i.e. sheared and yawed inflow, Grinderslev et al. (2021) showed that little differences are visible in the wake of a 2.3MW rotor using URANS in the incompressible Navier-Stokes solver EllipSys3D. Li et al. (2015) simulated the NREL 5MW rotor coupled to a multi-body dynamics solver and investigated local aerodynamic behavior under





turbulent inflow. Under these complex conditions they concluded that on the scale of the turbine diameter, turbulence has a larger influence on the wake of a rotor than the flexibility of blades. However, smaller scales in downstream wake regions closer to the rotor have not been investigated yet.

On the other hand, small scale numerical investigations of the rotor near wake expansions have been performed making use of Navier-Stokes actuator disc (AD), actuator line (AL) approaches and 3D vortex panel methods (Micallef et al. (2020)) or body fitted rigid Reynolds Averaged Navier-Stokes simulations (RANS) (Herráez et al. (2017), van Kuik et al. (2014)). Hodgkin et al. (2022) investigated the stability of a wind turbine tip vortex making use of large eddy simulations (LES) and an AL approach. Whilst AD/AL methods are limited to engineering corrections and small turbine RANS simulations assume the blades to be rigid, a coupled FSI investigation regarding the near wake expansion has not been performed according to the authors' knowledge.

Based on these findings, the objective of this study is to investigate the effects of blade flexibility and gravitational loading on the blade aerodynamic forces of an averaged sized MW turbine as well as their impact on the near wake behavior, i.e. wake velocity deficits and blade tip vortex trajectories. The rotor is investigated under rated operating conditions in uniform inflow perpendicular to the rotor plane. Although this setup is supposed to reflect optimal rotor blade behavior, we will show that in this baseline case already differences can be seen in the flow of rigid and flexible blades resulting in non-uniform wake behavior for flexible blades. It is assumed that these results are superimposed by even more influences under off-rated conditions. The aerodynamic findings in the rotor plane are linked towards near wake effects, highlighting the differences of the tip vortex trajectories and wake velocity deficits of rigid and flexible blades. With this, we explore the characteristic aerodynamic parameters that can be taken into account when performing more sophisticated flow scenario investigations like the above mentioned wind farm case.

In Sec. 2 the numerical setup including the solvers and post-processing methodologies is described. The results of the simulated cases focusing on aerodynamic characterization of rigid and flexible blades and turbine wake behavior are discussed in Sec. 3. Finally, a conclusion is given in Sec. 4, recapitulating the results and giving an outlook for future investigations.

## 2 Numerical methods and setup

In order to study the impact of structural flexibility on the aerodynamic behavior, simulations of the NREL 5MW generic reference wind turbine (Jonkman et al. (2009)) are performed. With a rotor diameter of 126m, it can be considered an averaged sized wind turbine in comparison to currently installed wind turbines (Energy (2021)). Additionally, several studies of coupled CFD-CSD simulations on that specific turbine were performed (Li et al. (2015), Yu et al. (2014), Liu et al. (2019)) that can be taken into account for comparison purposes. This is important, since for a generic wind turbine no measurement campaigns are conducted that simulations could be compared to.



## 2.1 Numerical discretization schemes and solver setup

Simulations are performed using the open-source software OpenFOAM (2022), where the incompressible, transient flow solver *pimpleDyMFoam* is coupled to the in-house non-linear structural finite element beam solver BeamFOAM (Dose (2018)). The structural beam implementation makes use of the finite element GEBT formulation that was originally proposed by Reissner (1972) and Simo (1985) and is capable of resolving large deformations of wind turbine blades. Within this study, 49 iso-parametric beam elements per blade are used corresponding to the generic NREL design, each consisting of two nodes with six degrees of freedom. Each beam accounts for gyroscopic effects and stiffening due to centrifugal forces. The temporal integration made use of the second-order generalized-$\alpha$ scheme (Arnold et al. (2007)). The coupling of fluid and structure is embedded inside the OpenFOAM framework without the necessity of external code communication and operates in a so-called loose coupling manner, which means that information is exchanged once per time step.

The incompressible, transient flow is simulated using the hybrid Spalart-Allmaras delayed detached eddy simulation method (Spalart et al. (2006)). To advance the solution in time, a second-order implicit backward method was used. The spatial discretization makes use of a second-order accurate Gauss linear scheme for the gradient terms and a first order Gauss upwind scheme for the divergence terms. The rotation of the blades is accounted for by using sliding mesh interfaces between the three blade grids, the rotor disk and the farfield grid.

## 2.2 Case setup

This work focuses on the blade flexibility, therefore the rotor is simulated without taking into account turbine specific effects like rotor-tower interaction, rotor-nacelle interaction or asymmetric behavior due to inflow shear, blade cone or rotor tilt. The rotor is simulated without the tower at rated conditions with a uniform inflow perpendicular to the rotor plane. The settings are listed in Table 1.

| Parameter | Value |
|---|---|
| Rated aerodynamic power | 5.3 MW |
| Number of rotor blades | 3 |
| Blade length / rotor diameter | 61.5 m / 126 m |
| Rated wind speed ($U_\infty$) | 11.4 m/s |
| Rated rotational speed | 12.1 RPM |
| Blade pre-cone angle | 0° |
| Rotor tilt angle | 0° |
| Blade mass | 17740 kg |

**Table 1.** Overview of the main rotor characteristics of the NREL 5MW reference turbine and the investigated conditions.





In order to investigate the influence of flexibility of the blades on the aerodynamic behavior and near wake effects, three simulations are performed: one with rigid blades (called *rigid* case in the following), one with flexible blades (called *flexible* case), and one with flexible blades excluding gravity effects (*flexible-noG* case).

## 2.3 Numerical Grid

The numerical grid for all simulations is based on the numerical grid used by Dose et al. (2018). They showed that a reasonable convergence for investigating aerodynamic effects taking into account fluid-structure interactions was reached for a mesh consisting of 36.38 million cells. This mesh was called the *Medium* sized mesh and is composed of five mesh parts, namely three blade meshes, a farfield and a rotor mesh. The domain measures 5 rotor diameters (D) upstream and 15D downstream of the rotor plane. All sides are located 3.5D from the rotational center. The overall domain set-up was kept, but as this study is highlighting the behavior of the tip vortex trajectory, additional mesh refinement was introduced in the wake of the blade tips to ensure a proper resolution of the flow in that region. As shown in literature (van Kuik et al. (2014), Micallef et al. (2016), Herráez et al. (2017), Micallef et al. (2020)), the trajectory of the tip vortex is expected to have a characteristic inboard motion in the order of 1%-2% of blade length. This movement needs to be captured properly. With the level of refinement used in this study, we ensure a minimum of seven cells within this scale. The highest resolution is located around the blade tip, where the vortex and blade surface are closest. Since the trajectory of the tip vortex is part of both the rotor mesh and the farfield mesh, both grid regions are equipped with an increased amount of cells, leading to a total grid size of 73.05 mio. cells. This already doubles the numerical costs of the *Medium* case mesh. The total amount of cells and their distribution after the refinements is shown in Tab. 2. In contrast to Dose et al. (2018), the setup used here allows very detailed vortex core tracking in the vicinity of the blade tip. An even higher cell refinement would exceed 100 mio. cells and would not be manageable by the used infrastructure.

| Mesh | Blade mesh cells $\times 10^6$ | Rotor mesh cells $\times 10^6$ | Farfield mesh cells $\times 10^6$ | Total cells $\times 10^6$ |
|---|---|---|---|---|
| *Medium* [Dose et al. (2018)] | 3 x 3.56 | 7.23 | 18.47 | 36.38 |
| *Tip refined* mesh used in this study | 3 x 3.56 | 13.11 | 59.93 | 73.05 |

**Table 2.** Amount of cells in different mesh regions, comparing the baseline *Medium* mesh of the former study described in Dose et al. (2018) with the *Tip refined* mesh used within this work.

## 2.4 Mesh deformation algorithm

To account for blade deformations inside the CFD domain, a mesh update algorithm developed by Dose (2018) was used. This method aims at providing a cost efficient and robust algorithm specifically developed for wind turbine simulations in OpenFOAM. Here, new mesh point coordinates are calculated in each time step without the need of solving a set of equations,





wherefore the deformation calculation costs are similar to those of a rigid mesh rotation. Within this method, all mesh volume cells surrounding the blade, as well as the blade surface cells are projected onto the moving blade beam using a Newton algorithm. Each cell is assigned to a specific beam element to account for the respective blade deformation magnitude. All mesh

cells are assigned into one of three mesh regions that are differently affected by the blade deformation. The closest region surrounding the blade surface moves rigidly with the blade, making sure no boundary layer cells are deformed during the process. Between the rigidly moved mesh region and the outer fixed zone, a polynomial of selectable order smooths out the blade mesh motion. Within this study, a first order polynomial is used. All cells outside of this transition region are not affected by any deformation.

## 2.5    Aerodynamic data extraction

The wake induction is extracted from the CFD simulation using the so-called 3-point method first introduced by Rahimi et al. (2016). Other aerodynamic quantities, i.e. AoA, lift and drag coefficients, on the rotor blades are derived from the calculated pressure distribution and the obtained induction factors. This method uses three points distributed along the chord on the

pressure and suction side of a particular section. These points are used to interpolate a representative AoA and local induced velocity while excluding effects of bound circulation as well as reducing the up and downwash of the blade. This method gives the possibility to account for dynamic effects like yaw misalignment, different azimuth positions and deforming blades also close to the blade root and tip, which is considered important for the investigation of blade tip vortex trajectories. Since the aerodynamic quantities are based on probing locations inside the fluid domain, the results are influenced by vortices trailing

from the blade surface, therefore the resulting quantities at the tip are to be considered a qualitative estimate to assess the aerodynamic behavior.

## 2.6    Simulations framework

Simulations are performed on the EDDY high performance computing cluster of the University of Oldenburg (HPC (2022)). 480 computing cores are used in each simulation run, computing a total of 220 seconds, i.e., 44 full rotor rotations. One

simulation run took about 25 days, with less than 1% more computational effort for the *flexible* case setup in comparison towards the *rigid* calculation. Simulation results were extracted from the last full rotor rotation if not stated differently.

## 3    Results and discussion

In the following, the three simulation cases, i.e., *rigid*, *flexible* and *flexible-noG* are investigated in more detail, highlighting their respective impact on the aerodynamic rotor blades behavior and connects these findings with the characteristics of the flow

in the wake of the rotor. The effect of flexibility and gravitation on the aerodynamic performance at the rotor blades is studied in Sec. 3.1. The resulting effects on the near wake, i.e. velocity induction and tip vortex trajectory, are shown in Sec. 3.2 and 3.3.



## 3.1 Aerodynamic behavior

Firstly, the overall global thrust $T$ and power $P$ quantities are computed by $\langle T(\Theta) \rangle_{rtn}$ and $\langle P(\Theta) \rangle_{rtn}$, where $\Theta$ describes the
azimuth angle and $\langle \rangle_{rtn}$ denotes the average over the last rotor revolution $0° \leq \Theta < 360°$, which corresponds to a time span of
$\Delta t = 4.959$ seconds. $0°$ and $360°$ define a blade pointing in upwards direction. The averaged quantities for the three simulation
cases are in good agreement with literature. Tab. 3 shows a comparison to various sources for rigid and flexible simulations of
the NREL5MW rotor, taking into account slightly different configurations with respect to rotor tilt, pre-cone or tower inclusion.
In the current investigation, the total thrust shows slightly higher loads when including blade flexibility compared to the rigid
case, while the total aerodynamic power slightly decreases. This is also confirmed in the study by Dose et al. (2018), which
uses a similar setup.

| Simulation | Thrust [kN] | Power [MW] |
|---|---|---|
| **Current work** [*rigid*] | 757.8 | 5.45 |
| Dose et al. (2018) [*rigid*] | 761.7 | 5.51 |
| Li et al. (2015) [*rigid*]* | 758.7 | 5.41 |
| Imiela et al. (2015) [*rigid*]** | 780 | 5.54 |
| **Current work** [*flexible*] | 759.2 | 5.43 |
| **Current work** [*flexible-noG*] | 760.0 | 5.44 |
| Dose et al. (2018) [*flexible-noG*] | 771.3 | 5.49 |
| Imiela et al. (2015) [*flexible*]** | 808 | 5.66 |

**Table 3.** Rotationally averaged thrust and power results for rigid and flexible setups. [*simulation contains precone, tilt and tower; **simulations contain pre-cone]

In the next step, the rotationally averaged force distributions of axial and tangential forces are investigated, i.e. $\langle F_{ax}(r, \Theta) \rangle_{rtn}$
and $\langle F_{tan}(r, \Theta) \rangle_{rtn}$. Under flexible conditions and a uniform wind coming from the front, the blades tend to bend towards lower
aerodynamic forces in the outboard region of the rotor, where the largest deformations are apparent. This holds for spanwise
positions outboard of $\approx 47$m ($\approx 75\%$ relative span) as shown in Figure 1, where the rotationally averaged tangential force in
the upper graph and the rotationally averaged axial force in the lower graph are smaller for flexible blades, respectively. This is
important, since most of the torque, and therefore power, is extracted from the wind in the outer third of the rotor blades due to
the force magnitude, the corresponding lever arms and the larger swept area. It is also shown, that neglecting the gravitational
loads barely influences the mean aerodynamic loads, as the corresponding tangential and axial forces are only slightly lower
than the loads of the flexible blades including gravity. For spanwise positions between 50% and 75% radius, the blade forces
including flexibility are slightly larger than for the *rigid* case, which aligns with the study by Dose et al. (2018). This is due
to the fact that the deformation of the blade in that region leads to slightly larger AoAs, in contrast to the behavior outboard of
75% span, where the AoA for flexible blades is smaller. Nevertheless, this part of the blade contributes less to the overall power



output, due to the smaller lever arm and smaller total difference. The blade region inboard of 50% blade span is not shown
since the total deformation in all investigated regions is much smaller and the resulting differences in the force distributions
are negligible.

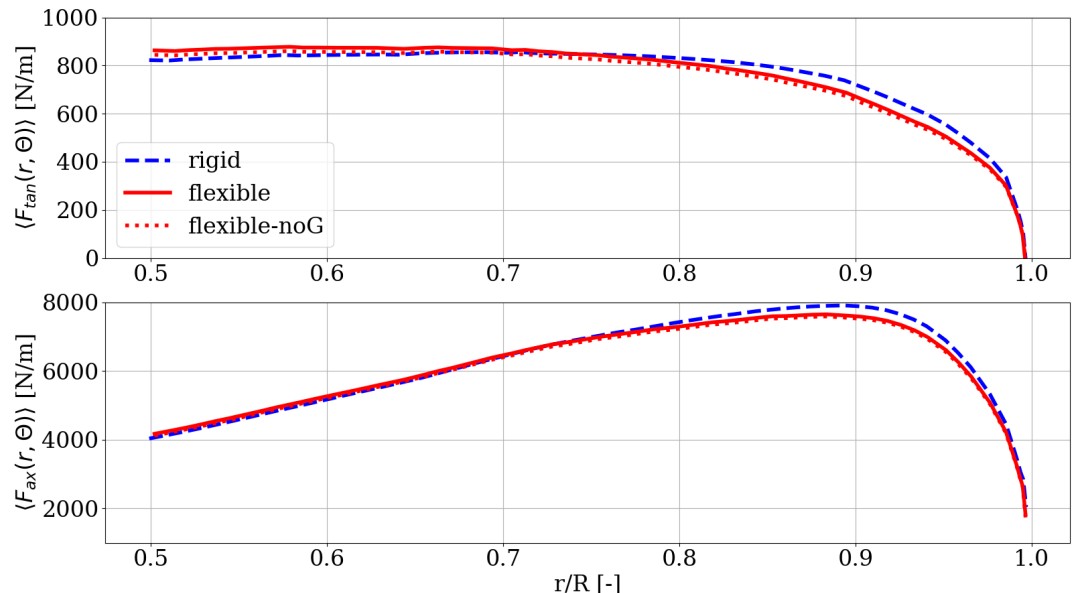

**Figure 1.** Rotationally averaged tangential and axial force distributions $\langle F_{tan}(r,\Theta)\rangle_{rotation}$ and $\langle F_{ax}(r,\Theta)\rangle_{rotation}$ for rigid and flexible
blades over the outer half of blade span related to normalized radial locations. $R$ is the rotor radius and $r$ defines the radial position on the
blade.

It is important to note that even though the curves of *flexible* and *flexible-noG* in Figure 1 lie almost on top of each other, a
detailed look into the distribution resolved over one rotation shows clear differences. For that purpose, the force development
of the 0.95 r/R location from Figure 1 is plotted against the respective azimuth angle in Figure 2. As visible in Figure 2(a),
the time series of the sectional forces at a spanwise position of 95% span shows almost constant forces for the *rigid* blade
and the *flexible-noG* case. While both flexible setups, consistent with the averaged forces of Figure 1, show lower force values
than the *rigid* case, a sinusoidal behavior within one rotation is clearly dominating the force development of the *flexible* blade
including gravity. The reduction of forces due to blade deformation aligns with the findings of Yu et al. (2014). The lowest
loads are given at an azimuth angle of around 90°, while the maximal loads occur around 270° azimuth. A frequency analysis
of the forces acting on the flexible blades (with gravity, i.e. *flexible* case, and without gravity, i.e. *flexible-noG* case) is shown
in Figure 2 (b). The dominant frequency in the *flexible* case is corresponding to the rotationally periodic 1P frequency (light
green vertical line), which is not existing in the *flexible-noG* case. However, both cases show a clear impact of the second
lowest eigenfrequency, which corresponds to the first edgewise mode (2nd vertical light red line). The larger impact of the
edgewise component of blade vibrations, in contrast to the flapwise mode, is based on the fact that for attached flows, the
aerodynamic damping in flapwise direction is significantly larger than in edgewise direction (Hansen (2007)) and is therefore





also more pronounced in the force frequencies. In Figure 2 (b) the 1P frequency and the other eigenfrequencies (Eigenf.) are clearly distinguishable.

While this frequency only has the second highest power spectral density in the *flexible* blade case, it dominates the force time series in the *flexible-noG* simulation. This clearly shows that the external 1P excitation only arises from the gravitation

and therefore is not present in the *flexible-noG* results.

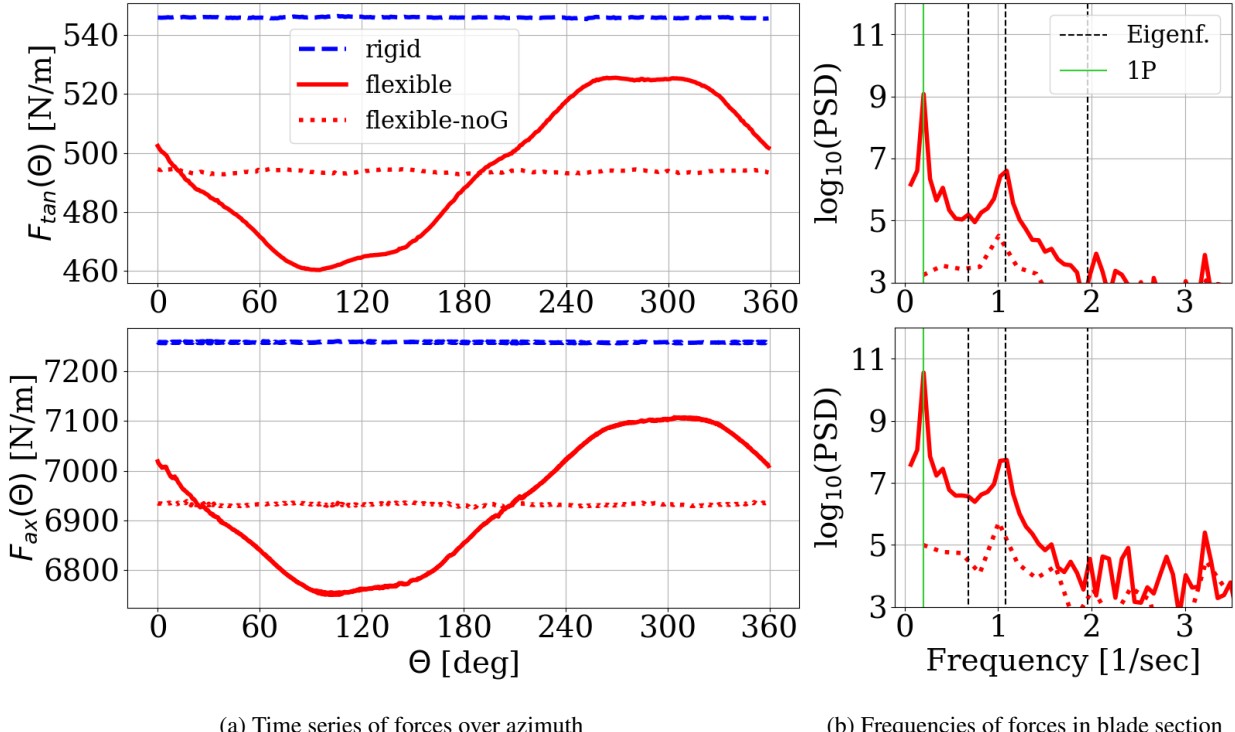

(a) Time series of forces over azimuth         (b) Frequencies of forces in blade section

**Figure 2.** Tangential force ($F_{tan}(\Theta)$) and axial force ($F_{ax}(\Theta)$) for the three cases of rigid and flexible blades over one rotation at 95% span (i.e. at $r/R = 0.95$), where the flexible blade forces are shown including gravity (solid red line) and without gravity (dotted red line). The rigid blade forces are shown in dashed blue.

In Figure 3, the blade tip deformation is shown, which also corresponds to the maximal deformation in edgewise, flapwise and torsional direction of the whole blade. Although being slightly further outboard than the section at 95% span from Figure 2, it describes an equivalent blade deformation behavior.

The dominating dynamic sinusoidal edgewise deflection component in Figure 3 (a) can be directly linked to the 1P gravi-

210 tational load in Figure 2, since the lever arm of gravitation for a blade in horizontal positions at 90° and 270° azimuth is the largest. The main cause of the reduction of power of the flexible blades in comparison to the rigid can on the one hand be traced back to the flapwise deflection (Figure 3 (b)), which results in a smaller area swept by the turbine blades and therefore less energy that is extracted from the wind. On the other hand, the torsional degree of freedom impacts the loading, since a lower



angle of attack is directly linked to a reduction of the local lift and drag components and therefore impacts the gliding ratio. As shown in Figure 3 (c), the torsional component is in phase with the edgewise deformation and has its largest influence at 90° and 270° azimuth.

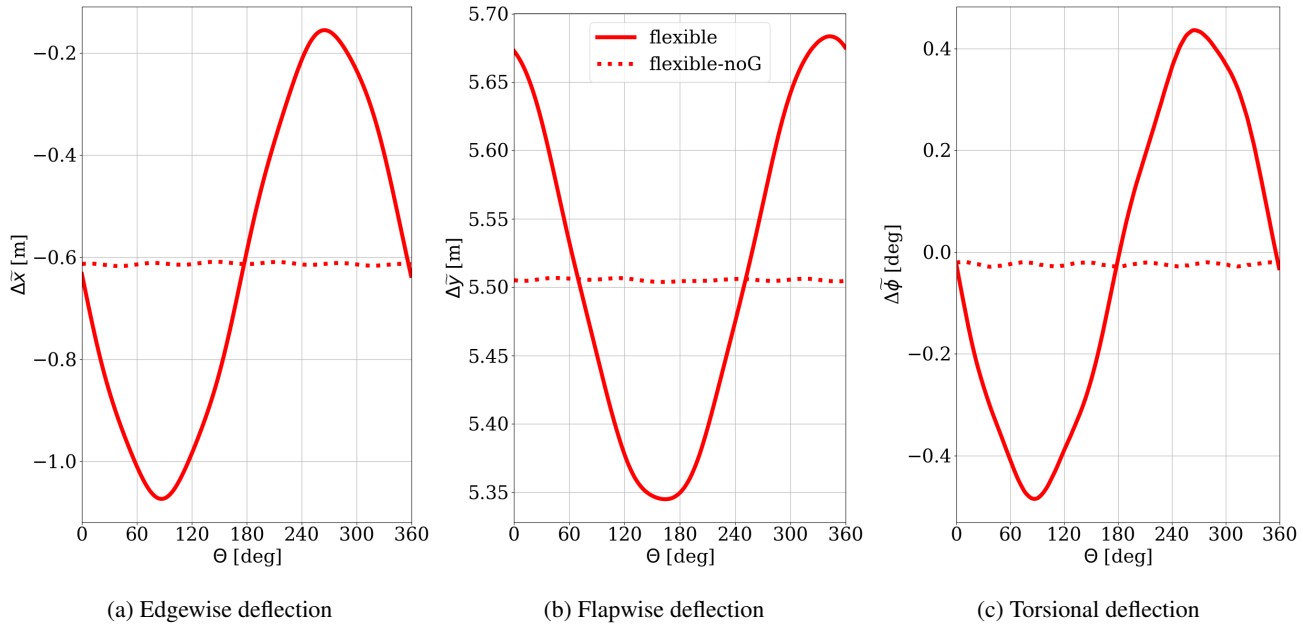

(a) Edgewise deflection     (b) Flapwise deflection     (c) Torsional deflection

**Figure 3.** Edgewise ($\Delta\widetilde{x}$), flapwise ($\Delta\widetilde{y}$) and torsional ($\Delta\widetilde{\phi}$) blade tip deformation over one rotation for the *flexible* and *flexible-noG* case at r/R=1.

This shows that a wind turbine rotor operating at rated conditions in a uniform inflow faces its largest influence of a dynamic excitation of structural deformation due to gravitational loads. Here, between $0°$ and $180°$ azimuth, the blade bends forward towards the leading edge and during the upward rotation between $180°$ and $360°$ azimuth, it bends backwards in the direction
of the trailing edge. Due to the structural nature of this blade, the center of mass (CM) is located downstream of the elastic axis (EA) on the chord line. This holds true for all sections along the blade, exemplarily shown in Figure 4. In combination with the local twist angles, the gravity force leads to a torsion of the blade. Thus, for a downward moving blade, the angle of attack (AoA) is reduced with increasing twist angle, while the AoA is increased for an upward moving blade, where the twist angle is reduced. This can be explained using the simplified velocity triangle, where the inflow speed $U_{inf} = U_\infty \cdot (1 - a)$
and the circumferential speed $U_{rot} = \omega r \cdot (1 + a')$ form the relative velocity $U_{rel}$. Here, $a$ reflects the local axial induction and $a'$ the local tangential induction factors. The constant $\omega$ gives the rotational frequency. This sinusoidal torsion behavior is superimposed by deformation due to the eigenfrequencies of the blade with a much smaller magnitude. Although barely visible in the three investigated deformation components (Figure 3), it can be observed in the resulting forces acting on the blade especially in the frequency domain (Figure 2(b)). In contrast, the *flexible-noG* case displays nearly constant aerodynamic
forces and deformation due to the fact that the gravitational induced loads are completely missing (Figure 3).





In the subsequent analysis of blade flexibility on different aerodynamic quantities, the focus lies therefore on the comparison of the rigid blades with the flexible blades including gravitational loads.

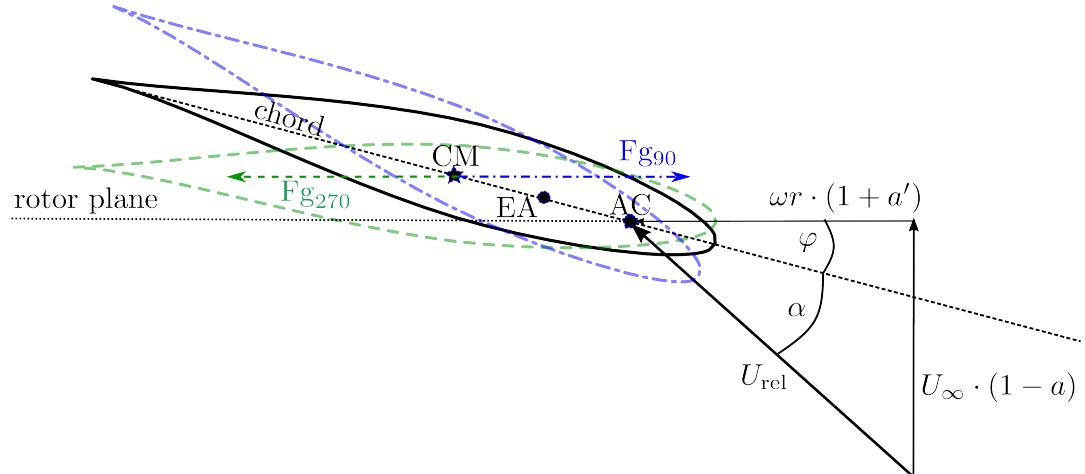

**Figure 4.** Simplified representation of the gravity force acting on a turbine blade section at radial position $r$. Black solid airfoil: undeformed blade twist condition of rigid blade, blue dashed dotted airfoil: nose down deformation for azimuth angles of $0° \leq \Theta \leq 180°$, green dashed airfoil: nose up deformation for azimuth angles of $180° \leq \Theta \leq 360°$. Fg$_{90}$ (in blue) and Fg$_{270}$ (in green) show the direction of gravitation for the respective azimuth angles. The local twist angle and AoA are given by $\varphi$ and $\alpha$, respectively. The relative velocity $U_{\text{rel}}$ results from the velocity components in axial direction ($U_\infty \cdot (1-a)$) and rotational velocity $\omega r \cdot (1+a')$, where $a$ and $a'$ define the induction factors in axial and tangential directions. $U_\infty$ defines the undisturbed freestream velocity and $\omega$ the angular velocity. Center of mass (CM), elastic axis (EA) and aerodynamic center (AC) are all located on the chord line.

### 3.1.1 Angle of Attack

As stated above, the inflow is constant and homogeneous for all cases, so the effect of the blade flexibility can be best assessed by focusing on the comparison between the rigid blades and the blades subjected to the periodic gravity-induced blade deformation. The *flexible-noG* simulations, in which no significant deformations occur (Figure 3), are therefore omitted from the analysis. Therefore, in the following all investigations refer to the *rigid* and *flexible* cases. A comparison of the local AoA for the most outboard 10% of the blade simulations (corresponding to $0.9 \leq r/R \leq 1$) is shown in Figure 5, where the respective azimuthal position is plotted against the radial location on the blade, showing *rigid* results in Figure 5a and *flexible* results in Figure 5b. The investigated AoA magnitude is accounted for with color bins of $0.4$ degrees.

For the *rigid* blade case, the AoA becomes larger with increasing radius but stays constant for each section independently of the azimuth angle. The radial increasing effect is also visible for the *flexible* blade, but in contrast to the *rigid* case, it deviates over the azimuth angle, showing the expected sinusoidal shape. The smallest AoA for the flexible blades is observed at $90°$ azimuth, whilst for $270°$ azimuth its maximum is present. This phenomenon can directly be linked to the torsional deformation,





as shown in Fig.3c, where the gravitation twists the blade towards lower AoA during a downwards movement of the blade and towards higher AoAs when moving upwards again.

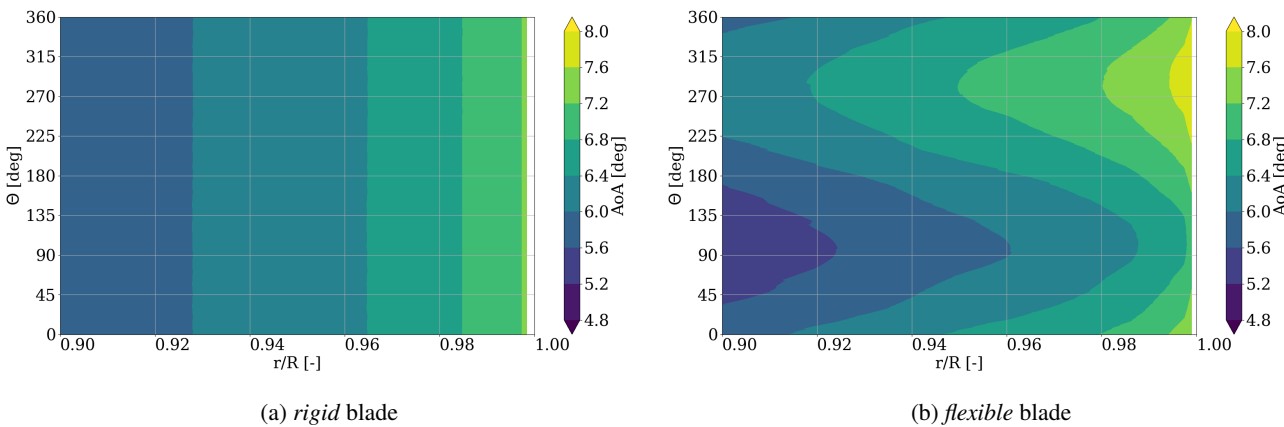

(a) *rigid* blade                    (b) *flexible* blade

**Figure 5.** Angle of Attack distribution for $0.9 \leq r/R \leq 1.0$ during one rotation for (a) rigid and (b) flexible blades.

### 3.1.2  Axial induction

Similar trends are visible in the representation of the axial induction (Figure 6), where each color bin represents an axial induction portion of 0.03. A general reduction of the axial induction towards the tip is visible for *rigid* and *flexible* blades. 250 At the most outboard position it vanishes due to the ever shrinking surface and the blade boundary layers merging between pressure and suction side of the blade.

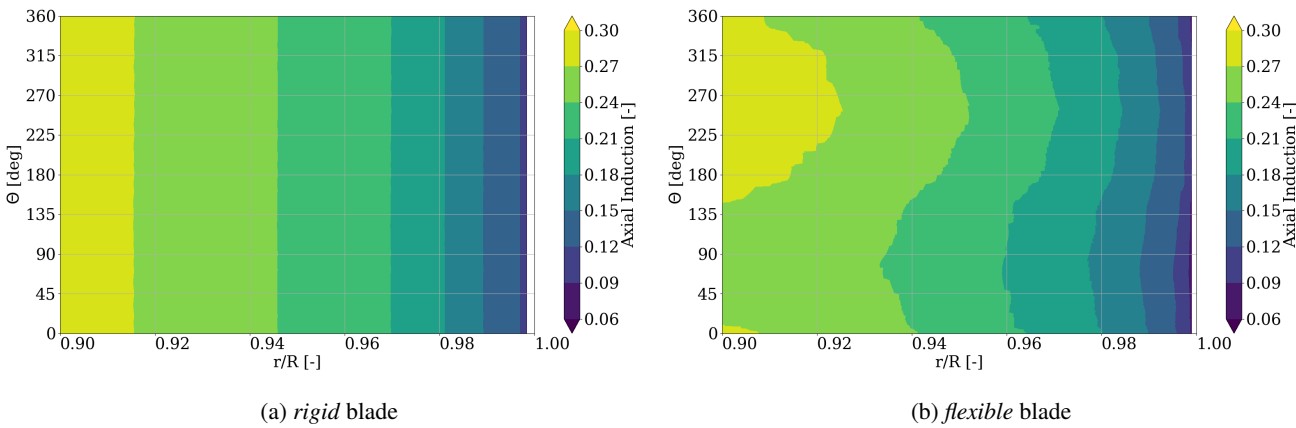

(a) *rigid* blade                    (b) *flexible* blade

**Figure 6.** Axial induction factor distribution for $0.9 \leq r/R \leq 1.0$ during one rotation for (a) *rigid* and (b) *flexible* blades.

Since the axial induction factor describes the amount of velocity reduction of the freestream normal to the rotor plane, it gives an estimate of the amount of energy taken from the fluid. From Figure 6, it is therefore expected that more energy is





extracted with the *flexible* blade pointing towards 270° than at 90° azimuth. This matches the findings in Figure 2, where the
maximal axial and tangential force is occurring at $\approx 270°$ and the minimal force is to be found at $\approx 90°$ azimuth, respectively.

### 3.1.3   Lift, drag and gliding ratio

Next to the forces described in Sec. 3.1, the lift, drag and gliding ratio coefficients play an important role, since they represent
the loading on the blades and allow for an estimate whether the blade is operating under optimal conditions. Since modern wind
turbines are lift driven, a maximal energy extraction from the wind is present, when the gliding ratio reaches its maximum,
which corresponds to the maximal ratio of lift over drag. Lift and drag are also direct inputs to engineering models like the
blade element momentum theory, so that a deep understanding of those quantities is vital for the improvement of such design
tools (Avatar (2018)).

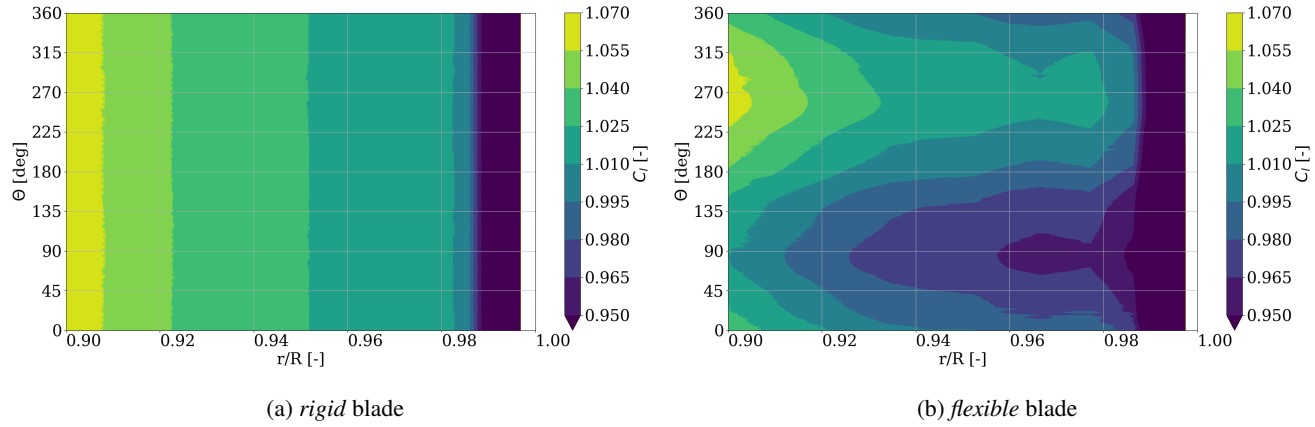

(a) *rigid* blade                                                      (b) *flexible* blade

**Figure 7.** Lift coefficient ($C_l$) distribution for $0.9 \leq r/R \leq 1.0$ during one rotation for (a) *rigid* and (b) *flexible* blades.

The coefficients for lift and drag are calculated from the simulation results using the local pressure distribution at various
sections along the blade leading to a force vector that is split into lift and drag directions using the angle of attack calculated
from the 3-point method in Sec. 3.1.1. Figure 7 shows the lift coefficient distribution of the *rigid* and *flexible* blade cases in
bins of 0.015, with consistent reduction of lift towards the tip. The minimal lift at 90° and maximal lift at 270° azimuth for
the flexible blades is as expected since the same behavior is shown in Figure 2 for the absolute force values in tangential and
normal direction.

The drag coefficient in Figure 8, contrary to the corresponding lift values, increases with radial distance. Since under these
operating conditions, the drag values in general are very small, drag bins of 0.0075 are shown. The distribution of minimal and
maximal drag coefficient values coincides with the findings of lift and AoA. This is an expected behavior, since most airfoils
operate in the so-called "linear region" when exposed to incoming angles of attack between 4.8° and 8.0° (cf. Figure 5), where
the boundary layer is fully attached to the blade surface and the drag force is directly proportional to the angle of attack.



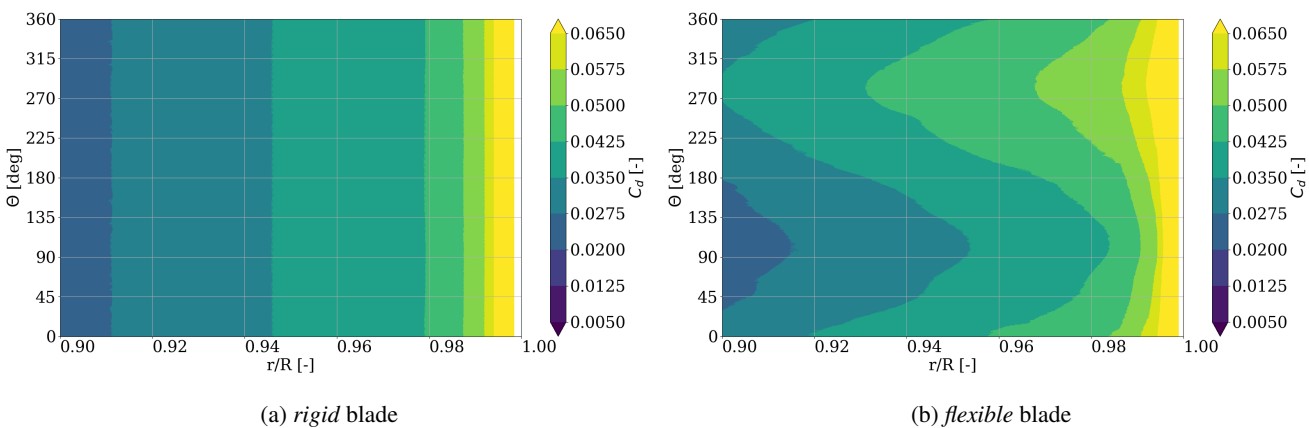

(a) *rigid* blade          (b) *flexible* blade

**Figure 8.** Drag coefficient ($C_d$) distribution for $0.9 \leq r/R \leq 1.0$ during one rotation for (a) *rigid* and (b) *flexible* blades.

Since in the investigated region the drag grows faster with increasing AoA compared to the lift, the gliding ratio of $Cl/Cd$
is not constant but shows sinusoidal behavior for the flexible blade over one rotation with a maximal gliding ratio for minimal
drag at an azimuth of $90°$ (Figure 9). From an aerodynamic standpoint, the blade airfoils in the investigated region therefor
show better performance when the blade is moving downwards, than for an upward moving blade.

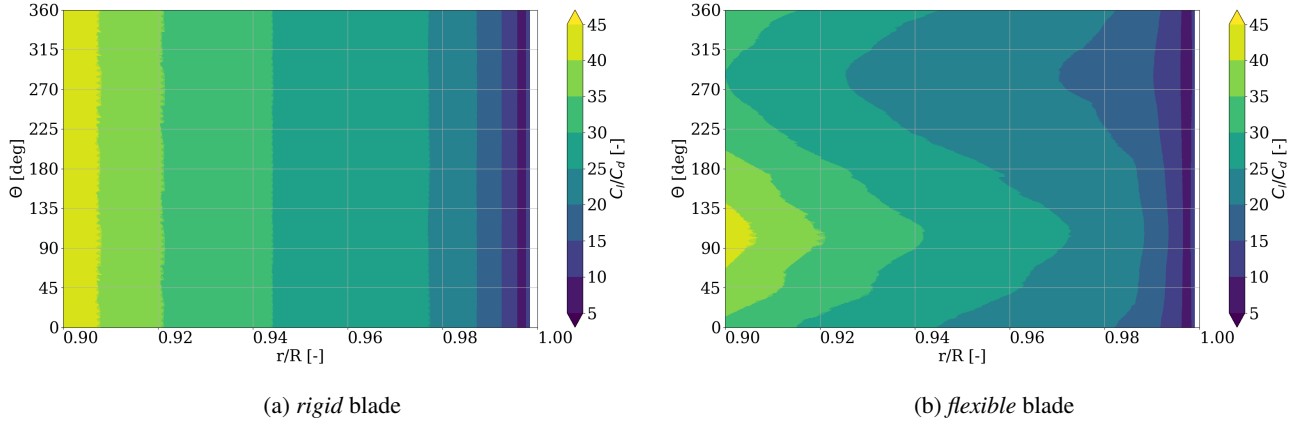

(a) *rigid* blade          (b) *flexible* blade

**Figure 9.** Glide ratio distribution for $0.9 \leq r/R \leq 1.0$ during one rotation for (a) *rigid* and (b) *flexible* blades.

Summarized, the aerodynamic investigation of the most outboard 10% of the blade (i.e. $0.9 \leq r/R \leq 1.0$) reveals a major
difference between *rigid* and *flexible* structures. The investigation of AoA, induction, and force coefficients shows a clear
dependency of the flexible blades on the azimuth angle. Since the aerodynamic behavior in the rotor plane characterizes the
interaction of blades with the surrounding air, differences in the induction due to flexibility are expected to be visible in the
wake of the turbine, too. This is studied in the next section.





## 3.2 Effects in the wake

To investigate the effect of a sinusoidal aerodynamic behavior within one rotation of the blade, a characterization of the
whole wake cross section at different downstream locations is made. Here, slices at 1/4D and 1/2D in the wake of the turbine
normal to the flow direction are investigated with respect to the mean speed magnitude for speed in $x, y, z$-direction, i.e.
$\left\langle \sqrt{u_x^2 + u_y^2 + u_z^2} \right\rangle_{rot}$. Here, $\langle\rangle_{rot}$ denotes the average over the last whole rotation. In Figure 10, the mean velocity distribution
of the wakes of both simulations are compared, with the *rigid* blade case in Figure 10(a) and the results for the *flexible* blades in
Figure 10(b). The velocity magnitude ranges are chosen such that the region of maximal blade deformation becomes apparent.
The *rigid* blade case shows a uniform wake velocity deficit over the whole azimuth with a maximal deficit at $\approx 80\%$ span. This
radial location of maximum speed deficit can also be observed in the case of the flexible blades but with a clear shift between
left and right, corresponding to $270°$ and $90°$ azimuth. Here, a velocity difference of $\approx 5\%$ is visible between the two sides.
Also the expansion of the visualized velocity deficit is larger at $270°$ than at $90°$ azimuth.

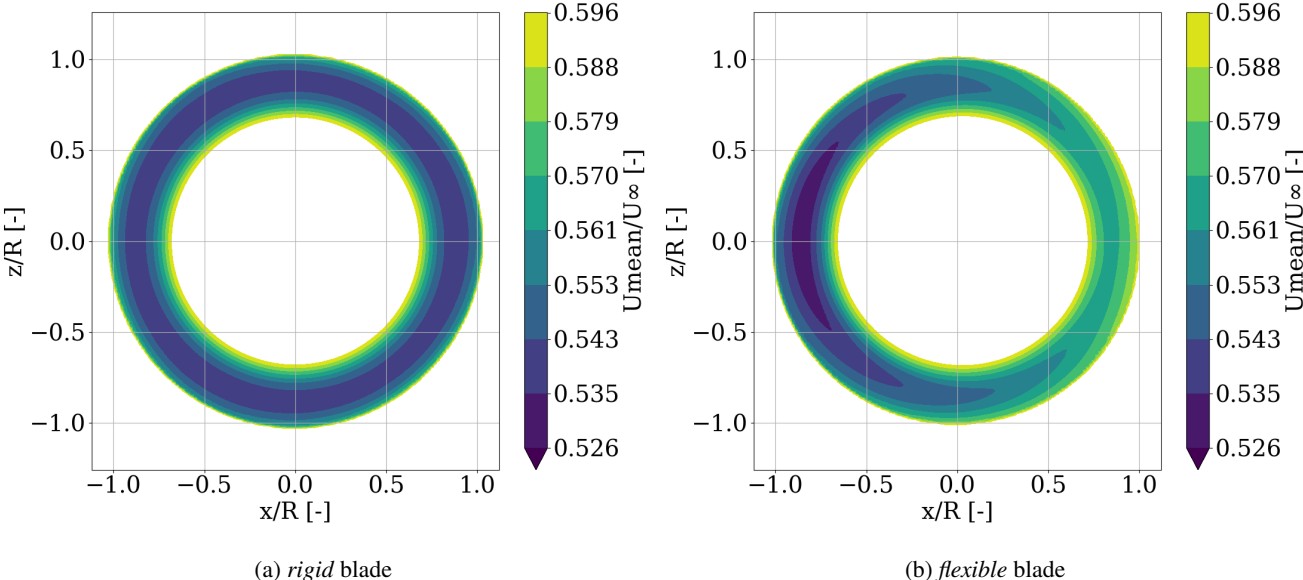

(a) *rigid* blade                    (b) *flexible* blade

**Figure 10.** Mean velocity magnitude slice in the $x$-$z$-plane at 1/4D downstream of the turbine with incoming flow direction in $y$-direction.
Color coding bounded such that all values of $U_{mean}/U_\infty > 0.596$ are shown in white, while all values of $U_{mean}/U_\infty < 0.526$ are marked
in dark blue.

Although less significant, similar results are shown for a 1/2D downstream location as shown in Figure 11. The total influ-
enced area seems to expand and smear out. This can be explained by the increasing mixture of fluid inside and outside the wake
stream tube compared to the 1/4D downstream location. The maximal speed reduction in the *flexible* case is shifted slightly
towards the ground (dark blue region), which is attributed to the counterclockwise rotation of the wake. This rotation opposing



the clockwise rotation of the turbine is based on the conservation of momentum and has been reported in many studies before (Zhang et al. (2012), Mühle et al. (2017), Shen et al. (2007)).

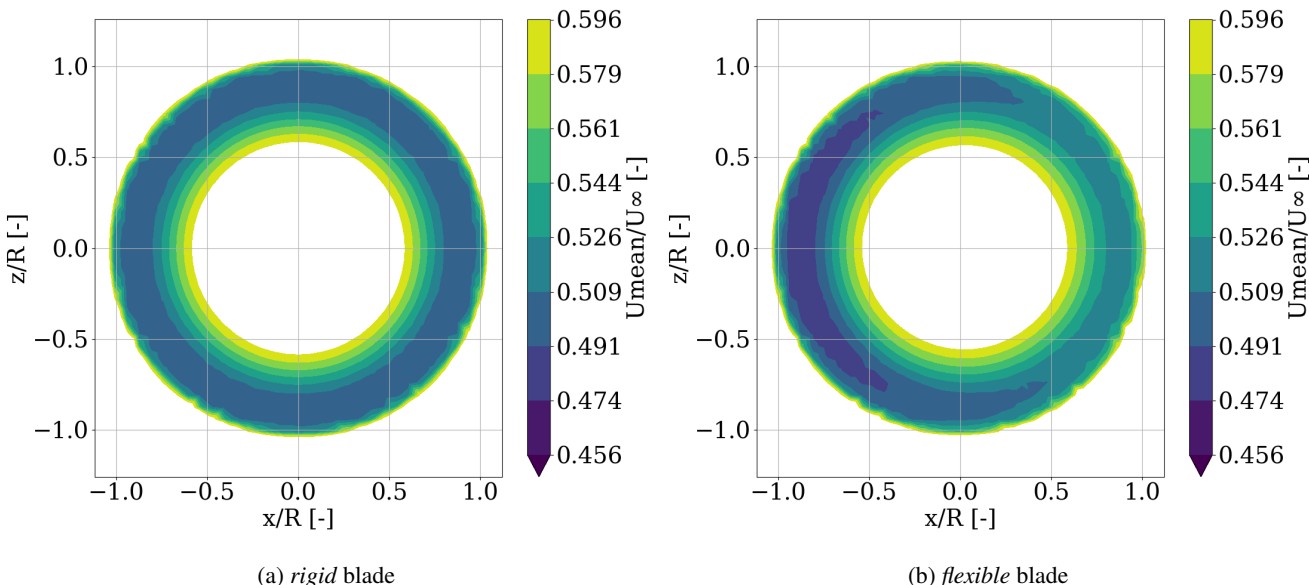

(a) *rigid* blade                    (b) *flexible* blade

**Figure 11.** Mean velocity magnitude slice in the $x$-$z$-plane at 1/2D downstream of the turbine with incoming flow direction in $y$-direction. Color coding bounded such that all values of $U_{mean}/U_\infty > 0.596$ are shown in white, while all values of $U_{mean}/U_\infty < 0.456$ are marked in dark blue.

This is consistent with the results from the aerodynamic investigation in Sec. 3.1, where higher velocity deficits in the wake of azimuthal positions of larger induction or higher angles of attack were observed. Thus, the impact of *flexible* rotor blades on the aerodynamic behavior of the wind turbine is as well transported into the wake and still visible at least 1/2D downstream of the turbine.

### 3.3  Tip Vortex Trajectory in the near wake

From the findings of the aerodynamic behavior in Sec. 3.1 and the mean wind speeds analyzed in Sec. 3.2, a clear difference between azimuth positions of 90° and 270° is evident for *flexible* blades in contrast to the *rigid* blade representation. Apart from the difference in mean wind speeds, also the shape of the wake stream tube deviates. To quantify these variations, a deeper investigation of the blade tip vortex trajectories is made in the following.

In general, the stream tube, being formed by all particles of the fluid moving through the rotor plane, has a smaller diameter
upstream of the turbine, where the speed is higher, than downstream of the rotor disk where the speed is smaller (Figure 12(a)). The larger the reduction of wind velocity, the wider the stream tube is to be expected further downstream as a consequence of the mass conservation. A representative structure, that describes the behavior of the stream tube at the blade tip, is the trajectory of the tip vortex of the blades, since it reflects the most outboard part of the fluid moving through the rotor plane.



Based on the comparison of the *rigid* and *flexible* blades in Sec. 3.2, different behavior for the wake expansion has to be
visible at different azimuth positions. For that purpose, the tip vortex trajectories are extracted from the flow field in horizontal
planes parallel to the inflow direction making use of the $\lambda_2$ criterion described in Jeong et al. (1995). Figure 12(b) shows a
representative $\lambda_2$ field extracted from the *rigid* blade CFD simulation at different time instants orthogonal to the rotor plane.
Here, $r/R$ defines the radial and $y/R$ the axial position normalized by the blade length. The wake age is defined such that
at $0°$ the blade tip is located at $r/R = 1.0$ and $x/R = 0.0$. During rotation, the blade passes through the slice, forming a tip
vortex that is transported downstream in time, following a characteristic path. For the uniform conditions of the *rigid* setup,
this path is independent of the blade position and can be extracted in any plane perpendicular to the rotor plane. To quantify
the influence of blade flexibility on the path, the vortex trajectory is analysed for tip vortices trailed at $90°$ and $270°$ for the
*rigid* and *flexible* simulation, since the largest deviations are to be expected at these positions. This radial location of the vortex
trajectory is defined by the position of maximal values of $\lambda_2$ inside the vortex core.

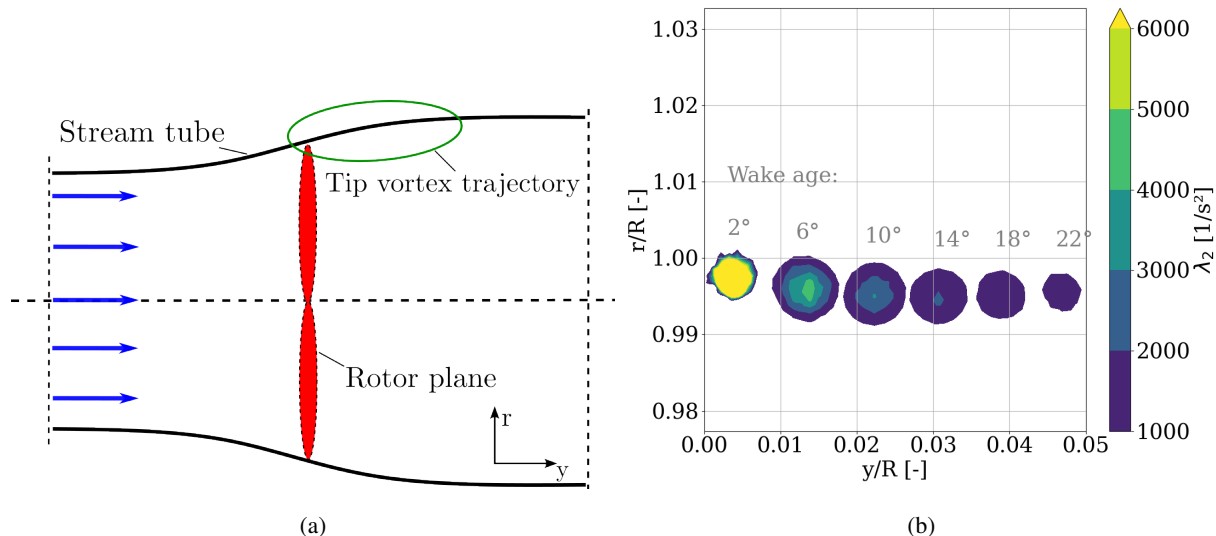

(a)                                                                                      (b)

**Figure 12.** Stream tube extraction from the $\lambda_2$ field: (a) general representation of a stream tube through a rotor plane. (b) tip vortex cores
at different instants in time. Color coding bounded such that all values of $\lambda_2 < 1000$ are shown in white, while all values of $\lambda_2 \geq 6000$ are
marked in yellow.

Figure 13 shows the non-dimensionalized radial location of that tip vortex trajectory averaged over the latest five times the
blade is passing through the plane of interest. The wake age is limited to $120°$, since this corresponds to the instant the next
blade is slicing through the plane, forming a new vortex trajectory heading downstream normal to the rotor plane. The radial
position $r$ of the vortex core is normalized by the blade radius $R$. Vortex core locations are tracked at each $2°$ of the wake age,
making sure that the near wake expansion is properly resolved. A characteristic inboard tip vortex motion is visible for all cases
for the first $20°$ wake age before expanding the stream tube and exceeding the actual radius of the rotor blades between $35°$
and $50°$ wake age. This characteristic trajectory aligns with the findings of van Kuik et al. (2014) and Herráez et al. (2017)





for their investigation for rigid blades, but a comparison between rigid and flexible blades is displayed for the first time in this work.

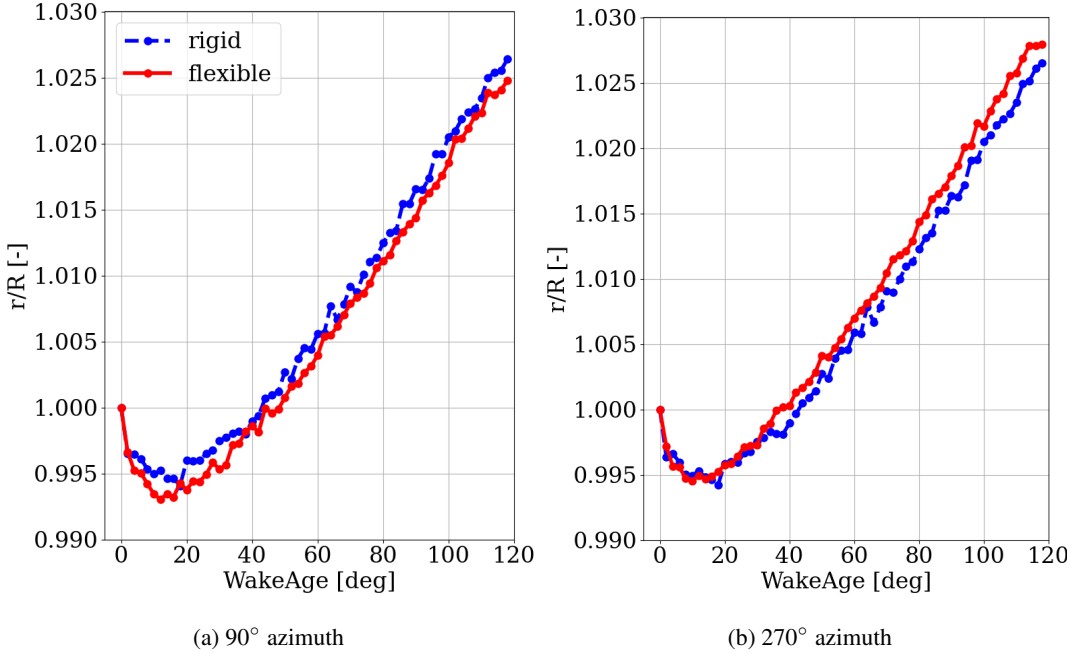

(a) 90° azimuth                    (b) 270° azimuth

**Figure 13.** Tip vortex trajectory for rigid blades (blue) and flexible blades (red) at 90° azimuth and 270° azimuth (averaged over five blade vortex trajectories).

It is visible that the radial location of the tip vortex of the rigid blade case exceeds the trajectory of the flexible blades at 90°
azimuth in the whole investigated region (Fig.13 (a)), whereas opposing behavior is shown at 270° azimuth (Figure 13 (b)). A
larger stream tube expansion corresponds to a higher velocity reduction and vice versa. This is supported by the findings made
in the previous section 3.1, where the investigation of axial induction factors in Figure 6 showed that the flexible blades induce
more velocity at 270° and induce less velocity in the axial component at 90° azimuth compared to the rigid blade case.

## 4   Conclusion

In the current study, three different simulations of the NREL5MW turbine were performed using a uniform inflow velocity
at rated conditions. A rigid blade setup is compared to a setup with flexible blades with gravity and flexible blades without
gravity.

It was shown that the aerodynamic forces of flexible rotor blades vary with the azimuth angle of each blade respectively.
This effect becomes stronger in outer parts of the blades, which contribute most to the global power output due to larger lever
arms, as a consequence of the deformations induced by the gravitational loads. In uniform inflow, this effect is dominated by

gravitational loading, that especially increases torsional deformations and, therefore, leads to a dynamic change in induction and AoA. Since the direction of gravitation strictly follows a 1P frequency, the resulting excitation of the blades shows a rotationally periodic sinusoidal behavior in all investigated aerodynamic quantities. With this turbine setup, the investigation reveals a much larger glide ratio for a blade moving downwards than for an upward moving blade, which leads to the conclusion that the investigated blade region performs better when the blade is descending.

Furthermore, it was shown that in contrast to the uniform wake of a *rigid* rotor, this varying aerodynamic behavior of the *flexible* setup is also transported into the wake, visible in rotationally periodic wake speed deficits. This deficit is rotating opposing to the turbine in counterclockwise direction leading to an asymmetry in the speed distribution varying with downstream distance. A 5% speed difference was observed for the *flexible* simulation case within one rotation at 1/4D and 1/2D downstream locations. This asymmetry, visible in the deformation of the wake shape, can be quantified by the tip vortex trajectory at 90° and 270° azimuth within the first 120° of wake age. This led to the conclusion that the radial near wake expansion downstream of the downward moving blades is smaller for the *flexible* blades than for the *rigid* blade setup. Opposing behavior was observed in the near wake of the blades moving upward. To the author's knowledge this effect has not been addressed before and it can serve as a basis for the analysis of more complex operating conditions.

With wind turbines becoming larger and larger, the described influences are expected to be more pronounced, since larger diameters are directly accompanied with more flexible structures. It is therefore worth studying these effects in more detail. Thus, these effects should be taken into account for future investigations and designs of large, flexible rotor blades, since the sinusoidal varying aerodynamic forces increase the impact of alternating blade loads on various turbine components. Also during conditions, where blades operate close to stall, the deformation due to gravitational loading could potentially cause unwanted flow separation and lead to highly dynamic phenomena, e.g dynamic stall. Additionally the decay of the wake further downstream and the impact of blade-flexibility driven skewed wakes on a following turbine, for example in a wind farm, should be investigated.

*Data availability.* The raw data of the simulation results can be provided by contacting the corresponding author.

*Author contributions.* Leo Höning performed the simulations, post-processing and analysis. Laura J. Lukassen and Iván Herráez provided assistance and guidance of the underlying methods. Leo Höning wrote the paper with corrections of Laura J. Lukassen, Bernhard Stoevesandt and Iván Herráez.

*Competing interests.* The contact author has declared that none of the authors has any competing interests.





*Acknowledgements.* The computations were performed on the high-performance computing system EDDY of the University of Oldenburg, which is part of the project WIMS- Cluster (FKZ 0324005) funded by the Federal Ministry of Economic Affairs and Climate Action.



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
