# Peer review of "Influence of rotor blade flexibility on the near wake behavior of the NREL 5MW wind turbine"

_Wind Energy Science, 2023_

## Referee Comment (RC2)

[referee-annotated manuscript omitted]

---

## Author Response (AR1)

We would like to thank you for your constructive feedback.

In the following, we would like to address your comments and suggestions individually.

RC1:

*Lines 50-54: It is unclear whether the authors are referring to the centrifugal forces, the gravitational forces, or something else. Please clarify.*

AC:

The sentences were changed, to clarify that the gravitational forces are described: "Despite its influence on deformation and loading, Sayed (2019) concluded that the gravitational impact on the total power output of a generic 10 MW wind turbine is negligible. The influence of gravitation on the turbine wake has not been investigated in detail yet but it is important to understand the influence of blade flexibility in the context of wind farm layouts, where the wake of a turbine in the first row affects the inflow of turbines in the second row."

RC1:

*Section 2.2. Since the work is focused on the behavior of the wake, additional information should be provided on the boundary conditions of the simulation, especially on the inflow conditions, as these could alter the wake response of the turbine significantly. Was a laminar inflow used? Otherwise, which values were used for the inflow turbulence?*

AC:

A short paragraph and a table with the provided values was inserted: "The boundary conditions of the rectangular domain are shown in Table 2. All sides are modelled as slip walls. The values of the turbulent quantities are set for laminar flow at the inlet. Fully turbulent conditions are assumed in the boundary layer on the blade surface.

RC1:

*Section 2.4. No information is provided about the discretization of the rotor region and boundary layer. The discretization of the blade is crucial for correctly capturing the aerodynamic response of the rotor and the behavior of the tip vortices. The authors cited the PhD thesis by Dose (2018), however I could not find it and it does not seem easily accessible for the interested reader. Nevertheless, further details should be included anyway, at least concerning the number of elements used to discretize the boundary layer and the subsequent value of the y+.*

AC:

The mesh and y+ description is added in Section "2.3 Numerical Grid". Additionally Figure 1c shows the cells in the boundary layer.

RC1:

*Line 198: There are no vertical red lines in Figure 2 (b). I imagine the authors are referring to the black dashed lines in the same Figure. Please clarify if this is not the case, and modify the Figure if necessary.*

AC:

The text referred to the former version of the figure and is now adjusted accordingly.

RC1:

*Line 226: I think the constant \omega expresses the rotational speed and not the rotational frequency.*

AC:

That is correct. The text was adjusted.

RC1:

*Line 226: "This sinusoidal torsion behavior is superimposed by deformation due to the eigenfrequencies of the blade with a much smaller magnitude. Although barely visible in the three investigated deformation components (Figure 3), it can be observed in the resulting forces acting on the blade especially in the frequency domain (Figure 2(b))." If I understand this correctly, the authors are pointing out the small-amplitude and high-frequency oscillations observed in the torsional deformation in Figure 3 (c). Then "barely visible" is referred indeed to such deformations. If this is the case, the authors should clarify this section of the text as it can be cause of confusion. Maybe it should be highlighted first that the torsional deformations described previously are clearly visible in Figure 3 (c). Then, it should be pointed out that there are also higher frequency components in the time-series which are due to the eigenfrequencies of the blade.*

AC:

This is indeed the case. We tried to point out that large amplitudes due to gravitation are superimposed by higher frequency oscillations due to the eigenfrequencies of the blade. The sentences have been rephrased to: "The dynamic torsion of the blade (Fig 3(c)) clearly shows the deformation due to the gravitation, forming a sinusoidal shape with a maximal deformation amplitude of |Delta phi|=0.5°. Additionally, higher frequency components can be found in the time-series and superimpose the sinusoidal torsion behavior with oscillations due to the eigenfrequencies of the blade with a much smaller magnitude. Although barely visible in the three investigated deformation components (Figure 3), it can be observed in the resulting forces acting on the blade especially in the frequency domain (Figure 2b))."

RC1:

*Line 5: change are to is.*

AC:

Done

RC1:

*Lines 29- 32: I believe the sintax of this sentence could be simplified for clarity.*

AC:

Changed to: "Yu (2014) showed that the aeroelastic deformation of the NREL 5MW wind turbine, which has a rotor diameter of 126m, results in a reduction of aerodynamic loads mainly due to torsional deformation. The non-linear Euler Bernoulli beam, that was used in that study, was coupled to their in-house incompressible fluid solver once per revolution."

RC1:

*Line 33: I believe a preposition and article are missing. The text should read "[…] based [on a] non linear […]"*

AC:

Done

RC1:

*Line 37: I believe the sentence could be improved: "This study showed that the averaged aerodynamic power output is reduced on the one hand, due to the blades bending towards a smaller rotor diameter and therefore less area to extract energy from, and on the other hand, [due to the] twisting [of] the blades towards lower angles of attack (AoA), which leads to a lower gliding ratio".*

AC:

Changed to: "This study showed that the averaged aerodynamic power output is reduced due to two reasons. On the one hand, the blades are bending towards a smaller rotor diameter, which results in less area to extract energy from, and on the other hand, due to the twisting of the blades towards lower angles of attack (AoA), which leads to a lower lift-to-drag ratio."

RC1:

*Throughout the text, Sect. should be used rather than  Sec., as per WES recommendation.*

AC:

Done

RC1:

*Lines 123 & 126: Two times the term "mio" appears, I believe referring to million. I think this might be a typo.*

AC:

Done

RC1:

*Line 276 typo: therefor*

AC:

Done

*Line 325: typo: "shows the non-dimensionalized radial location of [the] tip vortex trajectory […]"*

AC:

Done

*Line 336: The expression "the flexible blades induce more velocity at 270° and induce less velocity in the axial component at 90°" could be improved, as an increase in induction factor means a reduction in wake velocity and vice-versa.*

AC:

Changed to: "the flexible blades induce a higher velocity reduction at 270° and induce a lower velocity reduction in the axial component at 90° azimuth compared to the rigid blade case."
* * *
RC2:

l.53: the distance is at least a few diameters

and

l.60: the study focuses within 1/2D. There is a gap from the motivation highlighted previously.

AC:

We added a sentence explaining the importance of the fluid behavior close to the blade, as this will determine its movement downstream.

RC2:

l.66: Why FSI is necessary?

It might be necessary to discuss the new discovery, e.g., by comparing the results of this research to the listed literature in this paragraph.

AC:

To our knowledge an investigation of the tip vortex trajectory including FSI has not been performed yet. We conducted this analysis to find out if FSI has an impact on the behavior of the trajectory, which is indeed the case. We believe that a comparison towards the results from literature is not so relevant, as those results were generated for a turbine of much smaller scale (2m diameter), where blades are rigid in nature and FSI not relevant for those blades.

In Sect. "3.3 Tip Vortex Trajectory in the near wake", we link the inboard motion to results from literature, showing that the trajectory follows similar characteristics:

„A characteristic inboard tip vortex motion is visible for all cases for the first 20° wake age before expanding the stream tube and exceeding the actual radius of the rotor blades between 35° and 50° wake age. This characteristic trajectory aligns with the findings of van Kuik (2014) and Herráez (2017) for their investigation for rigid blades, but a comparison between rigid and flexible blades is displayed for the first time in this work."

RC2:

l.75: This is not discussed. Could you add it, e.g., to section 3.3?

AC:

We agree that this sentence sounds misleading. It has been rephrased to: "In this paper we investigate the characteristic aerodynamic parameters that are necessary for analyzing the near wake development. This can be of interest for wind farm studies."

RC2:

l.104: Please be explicit, tower, nacelle, shear, cone, tilt are not modelled. If nacelle is not modeled, how do you deal with it?

AC:

This paragraph is rephrased, to describe the used setup more explicitly. Since a nacelle is not used, the blades are extruded towards the center of rotation forming a merged structure of three cylinders. Additionally Figure 1b now shows the rotor: "This work focuses on the blade flexibility, therefore the complexity of the setup is reduced. The inflow is uniform and perpendicular to the rotor plane. The rotor is neither coned nor tilted, which leads to symmetric inflow conditions. The tower and nacelle are not modelled as well, and the blades are extruded at the blade roots towards the center of rotation to compose a closed hub region of the three cylindrical structures. Rotor operation is set towards rated conditions. All settings listed in Table 1."

RC2:

l.111: Could you add the mesh layout include the refinement region for repeatability of the work?

AC:

The mesh domain and refinement zone are inserted in Section "2.3 Numerical Grid", along with Figure 1 for clarification.

RC2:

What's the yplus?

AC:

The yplus description is added in Section "2.3 Numerical Grid" as: "The blade meshes were created using the BladeBlockMesher utility (Rahimi et al., 2016a), which composes 2D sectional meshes into a 3D structured blade volume mesh, consisting only hexahedral cells. The blade meshes are generated with a resolution of 260 cells in spanwise direction. The chordwise component contains 300 cells and 40 cells are distributed in blade normal direction. The cells normal to the surface follows a ratio of 1.2 and in order to limit the computational costs and to circumvent high aspect ratio cells inside the boundary layer, an adaptive wall function is applied (Fig. 1c). This wall function is capable of blending automatically between a high-Re and a low-Re approach, depending on the local y+ value. For the majority of the cells inside the first layer, a y+ value between 30 and 70 is applied. In situations with low velocities close to the blades and smaller y+ values, the wall function is switched off automatically to increase accuracy for e.g. flow separation."

RC2:

What's the turbulence intensity?

AC:

The turbulence intensity is accounted for in the new paragraph of Sect. 2.2: "The boundary conditions of the rectangular domain are shown in Table 2. All sides are modelled as slip walls. The values of the turbulent quantities are set for laminar flow at the inlet. Fully turbulent conditions are assumed in the boundary layer on the blade surface."

RC2:

l.141: From the "Evaluation of different methods for determining the angle of attack on wind turbine blades with CFD results under axial inflow conditions", the 3P method is not the best one, what's the motivation behind this choice?

The uncertainty on the AoA and force determination using the method is not discussed through the paper.

AC:

The motivation behind this choice is that the 3-Point method is an in-house development and therefore is tested for other applications. Additionally, this study aims at a relative comparison of 2 simulation setups, where absolute values play a minor role. Within this investigation unsteady inflow conditions are present, as the blades deform periodically. The big advantage of the 3-Point method is that it is suitable for such conditions in contrast to many other methods.

In section "Aerodynamic data extraction", the following sentence describes that the results are to be considered as a qualitative estimate and discussions are always considered relative to the other simulation result: "Since the aerodynamic quantities are based on probing locations inside the fluid domain, the results are influenced by vortices trailing from the blade surface, therefore the resulting quantities at the tip are to be considered a qualitative estimate to assess the aerodynamic behavior for a relative comparison between different blade setups."

RC2:

l.143: explanation for the first appearance

AC:

Explanation is included now: "Other aerodynamic quantities, i.e. angle of attack (AoA), […]"

l.143: only pressure drag then, friction drag is not included? could you mention it explicitly?

AC:

Friction drag is included as well. The sentence was changed to: "Other aerodynamic quantities, i.e. angle of attack (AoA), lift and drag coefficients, on the rotor blades are derived from the obtained induction factors and the calculated pressure distribution, as well as the viscous forces."

l.144: Are these points changing positions as well when the blade deforming in the simulation?

AC:

A sentence was added here to clarify that the points do change its position, to keep the distance towards the local blade chord constant: "The distance of the points towards the local chord is kept constant, to account for blade deformation during the simulation."

l.144: since on the pressure and suction side, how is it possible along the chord?

AC:

Sentence changed for clarification: "This method uses six points, three points on the pressure and three points on the suction side, which are distributed parallel to the chord of each analyzed blade section."

l.145: which section?

AC:

The sentence was changed as shown in the comment before this one.

l.150:

AC:

No comment has been found here.

RC2:

l.173 (table): Could you include the models used for all these simulations?

AC:

The models are included now.

RC2:

l.177: Sanity check with line 184

AC:

This sentence has been adjusted to: "This is important, since most of the torque, and therefore power of wind turbines, is extracted from the wind in the outer third of the rotor blades due to the corresponding lever arms and the larger swept area." Also see the comment for l.184 below.

RC2:

l.181ff: Could you explain why AoA for flexible blades is smaller for >75% span?

If it's due to the deformation, 75% span should deform more than 50%-75% span, right? Then the AoA should be even higher for the flexible blade for this outboard region.

I'm asking is because that the method to extract the AoA and force is pure 2D, it's not sure if it's still accurate for this tip region due to the strong 3D flow caused by the deformation.

It's not clear yet the behavior at >75% span is physic or is due to uncertainty in the used method.

This is a very important point as most of the following discussion focuses on the tip region, this uncertainty needs to be quantified.

AC:

We have rechecked this section, the mentioned literature (Dose et. al 2018), and the results again and we have to rephrase the explanation of the force distribution: "This is due to the fact that the deformation of the blade leads to slightly higher AoAs for the flexible cases throughout the whole span. In the blade region outboard of 75% r/R the forces are lower for the flexible cases. This can be explained by two counter acting phenomena. On the one hand the higher angle of attack leads to higher loads on the blade. On the other hand, the large flapwise deformation leads to a reduction of the rotor radius and thus in a lower circumferential speed and a change in inflow angle closer to the tip. The contribution of the angle of attack on the loads can be considered approximately linear, whereas the influence of the speed on the blade forces is quadratic and therefore prevails. The resulting forces outboard of 75% are therefore smaller for the flexible blades although a larger AoA is present."

RC2:

l.184:

AC:

This sentence has been adjusted for clarification: "Nevertheless, the radial part between 50% and 75% of the blade contributes less to the overall power output than the outer part, due to the smaller lever arm. Also a smaller total force difference between the rigid and flexible case than in the most outboard 25% of blade span makes this region less attractive for this study, which aims at highlighting the differences between these setups. The blade region inboard of 50% blade span is not shown since the total deformation in all investigated regions is much smaller and the resulting differences in the force distributions are negligible."

RC2:

l.214: could you add the reference?

AC:

The reference "Computational fluid dynamics of windturbine blade at various angles of attack and low Reynolds number" was added.

RC2:

l.256: It's normally not called gliding ratio in wind energy community, right?

AC:

It was changed to "lift-to-drag ratio" throughout the text.

RC2:

l.293: Perhaps plot the velocity deficit instead of normalised velocity. Right now authors say larger deficit which is not larger value in color scale.

AC:

We changed expressions in the text from "largest velocity deficit" to "minimal speed". Now a „minimum" also corresponds to the smallest value in the plot.

RC2:

l.295: Sanity check required; with, for example,  https://doi.org/10.1016/0167-6105(88)90037-2.

Perhaps this region should not be influenced by faster wake mixing for standard rotors.

AC:

The mixing between inside and outside of the streamtube is most probably not the case, as described by Ainslie 1988. However, turbulent mixing inside the wake itself is present, as the rotor creates turbulence inside the streamtube. The paragraph is changed to: "Since mixture of fluid between the outer free-stream and the inner wake is not expected within 1 to 2 diameters downstream of the turbine (Ainslie1988), this can be explained by increasing turbulent mixture inside the stream tube, that is created by the rotor itself. This effect is more dominant at the 1/2D downstream location compared to the 1/4D downstream location."

RC2:

l.297: Counterclockwise rotation is there for both rigid and flexible cases, right?

AC:

Yes, this is true. The following sentence was added to account for your comment: "Similar rotation is also present in the rigid blade case, but it is not visible here due to the uniform character of the rigid rotor wake."

RC2:

l.307: some discussion on the shape needs to be added in previous section

AC:

We have added a discussion in the ring shape of the uniform case in contrast to the deformed ring shape of the flexible case in Sect. 3.2 "Effects in the wake".

RC2:

l.323: Could you add "References or Physical Explanation"? It's not clear yet.

AC:

The paragraph has been rephrased to base the necessity of these performed analysis on the results from the previous two sections: "The largest deviations of the aerodynamic quantities (Sect. 3.1) and the wake velocity reductions (Sect. 3.2) are present for the azimuth angles, where the blade is positioned horizontally, i.e., 90° and 270°. Therefore, it is also expected that the largest differences in the tip vortex trajectories are visible under these conditions. To quantify these differences and the influence of blade flexibility on the path, the vortex trajectory is analyzed for tip vortices trailed at 90° and 270° for the rigid and flexible simulation. This radial location of the vortex trajectory is defined by the position of maximal values of lambda2 inside the vortex core."

RC2:

l.333(figures): could you add the uncertainty bar for this plot between the five trajectories?

AC:

Errorbars, in terms of max and min values of the averaged trajectories, have been added to the plots.

RC2:

l.337: What could be the consequence of this on a rotor?

Could you elaborate it a bit?

AC:

Elaboration has been added with respect to the consequences on the non-uniform rotor loading: "Apart from the impact on the wake flow, this effects the loading of the rotor as well. Non-uniform blade root bending moments are the consequence for each of the three blades, leading to an additional 3P main shaft and tower top excitation in case of a full wind turbine setup. Consequently, this needs to be considered in the design process of large flexible rotors."

RC2:

l.339: Maybe add some quantification (difference between cases) and uncertainty and limitation of the methods in this section?

AC:

We have added the following explanation to the outlook section: "Aerodynamic quantities have been extracted from the flow field using the 3-Point method. As with any other method, uncertainties in the obtained results are unavoidable. However, the uncertainties apply in the same manner to the rigid and flexible blade cases. Since the focus of this investigation lies on a relative comparison between the rigid and flexible cases, and not on the absolute value of the aerodynamic quantities, the possible influence of the uncertainties is considered not to be critical for our analysis."

RC2:

l.401: author list is not complete, could you double-check for all?

AC:

All authors were checked, and the style was adjusted. Alphabetical sorting in the References list, as well as in text citations have been changed towards WES guidelines.

---

## Author Response (AR2)

We would like to thank you for your constructive feedback.

Please find below our answers to your comments.

RC1:

*Line 132: Typo: Consisting only [of] hexaedral cells*

AC:

Done

RC1:

*Lines 131-138. The description of the domain discretization was improved in the reviewed version of the manuscript. Considering that for most of the cells the value of the y+ is of the order of 30 and 70 a wall function is employed. Further details could be provided about the wall function. Was this strategy tested in previous work by the authors? Why was this methodology chosen? What are its limitations in the applied test case? Please provide a brief description of the points above.*

AC:

The investigation in this study utilizes a pre-validated blade mesh, which determines the local y+ values to the existing near body grid resolution. To maintain accuracy within the boundary layer, a wall function is employed, ensuring a consistent turbulent viscosity profile for all simulated walls. This chosen setup and methodology aim to minimize computational costs within the boundary layer while offering enhanced resolution for studying the near wake losses and tip vortex trajectories with utmost precision. In recent years, this strategy has been widely employed both within the institute and the OpenFOAM community. It is acknowledged that this approach has certain limitations, especially when dealing with highly separated flow scenarios such as non-operating conditions with high angles of attack. However, in the case of this study, the turbine is simulated under rated conditions, and the local blade sections operate within the linear range. As a result, these non-design conditions do not apply here. The mentioned paragraph has been rephrased to:

"[...] This wall function is capable of blending automatically between a high-Re and a low-Re approach, depending on the local y+ value. For the majority of the cells inside the first layer, a y+ value between 30 and 70 is applied. Here, the wall function ensures a consistent turbulent viscosity profile for all simulated walls. [...] "